Response of conventional sunflower cultivars to drift rates of synthetic auxin herbicides

Serim Ahmet Tansel ahmettansel.serim@bilecik.edu.tr 1 2
Patterson Eric L. 2
1 Department of Plant Protection, Bilecik Seyh Edebali University , Bilecik , Türkiye
2 Department of Plant, Soil and Microbial Sciences, Michigan State University , East Lansing , MI , United States of America
Tariq Mohsin
Electronic publication date: 2024 Jan 10
Publication date: 2024
Volume: 12
Electronic Location ID: e16729
Received 2023 Aug 2; Accepted 2023 Dec 6
Copyright: ©2024 Serim et al.
Copyright year: 2024
Copyright holder: Serim et al.
License: This is an open access article distributed under the terms of the Creative Commons Attribution License, which permits unrestricted use, distribution, reproduction and adaptation in any medium and for any purpose provided that it is properly attributed. For attribution, the original author(s), title, publication source (PeerJ) and either DOI or URL of the article must be cited.
License URL: https://creativecommons.org/licenses/by/4.0/

Keywords: Quinclorac, Florpyrauxifen-benzyl+penoxsulam, Sunflower, Crop injury, Drift

Funding: The authors received no funding for this work.

==============================
The agrochemical industry has launched several new synthetic auxin herbicides in rice to combat increasing numbers of herbicide resistant weeds to other modes of action. Excessive or inappropriate use of these herbicides has resulted in unintended consequences near the sites of application, such as herbicide drift. This study was conducted to determine the impact of drift of quinclorac and florpyrauxifen-benzyl+penoxsulam (FBP) on the yield and yield components of two sunflower cultivars. In a growth chamber experiment, quinclorac and FBP were applied to 2–4 true leaf stages at rates ranging from 2.93 to 93.75 and from 0.51 to 16.25 g ai ha−1, respectively. Nonlinear regression analyses indicated that the cultivar Bosfora was more sensitive to quinclorac and FBP than the cultivar Tunca. In field experiments, these sunflower cultivars were treated with drift rates of quinclorac (<375 g ai ha−1) and FBP (<65 g ai ha−1) when they were at the 8-10 true leaf stage. Quinclorac and FBP drift rates resulted in up to 52-61% and 85–100% injury and 82-88% and 100% yield loss, respectively. Crop injury and yield data clearly showed that cultivar Bosfora was more sensitive to FBP and quinclorac rates than cultivar Tunca, and both cultivars were more sensitive to FBP than quinclorac. In our work, we also found that plant height reduction caused by quinclorac at early growth stages may be a valuable indicator to evaluate crop injury and yield loss.

Introduction

Thrace, located on the western side of Turkey, is a prominent agricultural region and produces large quantities of wheat, canola, sunflower, rice, and grapes due to its fertile soils, water availability, and ideal climatic conditions. This region alone provided 44.5% of the domestic rice and 41.1% of Helianthus annuus L. (sunflower) production in 2021 (TUIK, 2022). Furthermore, the agricultural productivity of this region has created a rich agricultural industry processing these crops, including oil and feed factories, mills, and food companies. It should be noted that although this region produces a majority of the sunflower and rice that Turkey consumes, one-third of sunflower and one-fifth of rice consumed in Turkish markets are imported from abroad (TUIK, 2022). Moreover, Turkey ranked first among countries that import sunflower in 2021 (OEC, 2023). Under pressure to increase productivity, growers from this region increasingly rely on chemical means to control pests such as weeds and diseases and have reduced the numbers of fields in fallow. Due to the reduction in fallow fields, sunflower fields and rice paddies are often adjacent.

Chemical weed control via herbicides has several advantages for many farmers including rice farmers; it is economical, easy, and overall efficient. However, among weed species that are common in rice paddies, a number of biotypes have developed resistance to herbicides due to over-reliance on only a few herbicidal sites of action (SOA) (Altop et al., 2014; Haghnama & Mennan, 2020; Kacan et al., 2020). Herbicides are classified based on their SOA which is the specific protein where the herbicide binds and subsequently inhibits weed growth; this classification helps growers manage herbicide resistance. For instance, group 1 and 2 herbicides inhibit acetyl CoA carboxylase (ACCase) and acetolactate synthase (ALS), respectively, while group 4 herbicides like quinclorac act as synthetic auxin (Mallory-Smith & Retzinger, 2003). Farmers who were living in the Thrace Region of Turkey have increased rates of ALS and ACCase inhibiting herbicides due to the failure of recommended rates (Serim et al., 2020). To address the increasing weed resistance to the commonly used herbicide SOA, some pesticide producing companies launched alternatives to ALS and ACCase inhibiting herbicides, including synthetic auxins in 2019 (PPPD, 2022).

Many rice farmers have been making multiple applications by using auxinic rice herbicides, which were registered to control weeds in a single season to achieve effective weed control. Spot spraying in rice fields is often done using a knapsack mist blower. This practice of spot spraying late in the season has increased the off-target movement (OTM) risk of herbicides, such as synthetic auxins from rice fields to sunflower fields of the Thrace Region. Previous research indicates that the severity of the drift caused by herbicides may change depending on the herbicide molecule, rate of herbicide, application parameters, weather conditions, growth stage, and non-target crop species (Cederlund, 2017). Sunflower is considered one of the most sensitive crops to herbicide residues and drift, especially synthetic auxins, ALS and EPSP synthase inhibitors (Greenshields & Putt, 1958; Lanini & Carrithers, 2000; Serim & Maden, 2014; Serim, 2022). Wall (1996) estimated that 2.4-D amine can result in a 93–100% yield reduction when applied at 151.2 g ai ha−1(24% of the low recommended rate) in sunflower. Additionally, herbicide drift has threatened the sustainable use of herbicides in agricultural fields. For instance, recent studies have shown that sub-lethal rates of herbicides, such as those caused by OTM, may reduce the sensitivity of weeds to herbicides (Vieira et al., 2020).

Quinclorac and FPB are new herbicide active ingredients registered to control weeds in paddy fields of Turkey as of three years ago (PPPD, 2022). Quinclorac (3,7-dichloro-8-quinolinecarboxylic acid) is a Group 4 herbicide that is thought to have two modes of action: (1) it seems to prevent cell wall biosynthesis and increases ethylene and cyanide production in grassy weeds while (2) it acts to mimic native auxin when applied to broadleaves weeds (Weed Science Society of America, 2014). OTM of this herbicide can cause severe injury on sensitive crops that grow adjacent to fields where it is applied, especially broadleaf crops, including tomato, pepper, cotton, and tree species (Snipes, Street & Mueller, 1992; Lovelace et al., 2007; Adams et al., 2017; Kaya et al., 2023). Florpyrauxifen-benzyl is an auxin-type herbicide (Group 4) belonging to the arylpicolinate chemical family that is used to control weed species in rice fields (Weed Science Society of America, 2014). This herbicide molecule is unique because of its wide herbicidal spectrum (Miller & Norsworthy, 2018a). Previously developed auxin herbicides like quinclorac are used to control narrow-leaf grass weeds, whereas florpyrauxifen-benzyl also has the ability to control many broadleaf weed species. Furthermore, it is often used to kill weed biotypes resistant to ACCs (Group 1) and ALS (Group 2) inhibiting herbicides in paddy fields such as Echinocloa spp.

Several dicot crop species are sensitive to florpyrauxifen-benzyl. Miller & Norsworthy (2018b) note that soybean was the most sensitive to florpyrauxifen-benzyl in tested crops, and moderate injury was observed on the seedlings at 14 DAT at 1/100 of the recommended rate. Schwartz-Lazaro et al. (2017) stated that florpyrauxifen-benzyl applied to soybean resulted in 71 and 31% injury at 21 DAT when applied with 1/20 and 1/80 drift rates, respectively. Cotton and sunflower also show severe injury at the 1/10 rate of the herbicide at 14 DAT and the herbicidal impact was mitigated by 28 DAT, especially at the 1/100 and 1/500 rates (Miller & Norsworthy, 2018b). Grass crops such as grain sorghum and corn seem to be unaffected by the herbicide (Miller & Norsworthy, 2018b). Because of the sensitivity of some broadleaf crops, florpyrauxifen-benzyl should be used with mitigation measurements to prevent damage on broadleaf crops. Therefore, recommendations should include a buffer zone and/or the use of drift reduction nozzles to protect non-target plants in some countries (Arena et al., 2018).

Although some studies have investigated the impact of sub-lethal doses of previously available synthetic auxin herbicides on sunflower, the impact of florpyrauxifen-benzyl has not yet been studied in Thracian sunflower production. The OTM of synthetic auxins is not unique to Turkey and has the potential to be problematic in many countries where rice and sunflower fields are in close proximity such as Italy, Greece, Russia, Argentina, Spain, Ukraine, and Bulgaria. The aim of this study was to determine the impact of sub-lethal rates of florpyrauxifen-benzyl + penoxsulam and quinclorac used in rice fields on the growth and yield of sunflower cultivars.

Material and Methods

Growth chamber experiments

A bioassay study was conducted in a growth chamber adjusted to 25 ± 1 °C/20 ± 1 °C with a 12/12 h day/night photoperiod to determine the sensitivity level of two commercial sunflower cultivars, Bosfora (Bosfora®; Syngenta Corporation, İzmir, Turkey) and Tunca (Tunca®; Limagrain Corporation, Bursa, Turkey), to sub-lethal rates of florpyrauxifen-benzyl + penoxsulam (BAXIGA® 32.5 OD; Corteva Agriscience Corporation, Istanbul, Turkey) and quinclorac (Facet®; BASF Corporation, Istanbul, Turkey). The growth chamber was set light to HPI-T Plus Metal Halide lamps (400 W). Three to four sunflower seeds were sown in plastic pots (7 × 7 × 8.5 cm) filled with white peat bedding substrate (TS 1, Klassman-Deilmann GmbH consisting of TS 1 fine + 15% perlite). After emergence seedlings were removed so that only two plants remained per pot. A high-performance air conditioning system was used to acclimatize the growth chamber because of adjusting of high temperature released by metal halide lamps. Therefore, the pots were arranged in a randomized complete block design with four replications according to acclimatization.

In the experiment, the seedlings were treated with seven herbicide rates (0, 2.93, 5.86, 11.72, 23.44, 46.88, and 93.75 g ai ha−1 for quinclorac and 0, 0.51, 1.02, 2.03, 4.06, 8.13, and 16.25 g ai ha−1 for florpyrauxifen-benzyl + penoxsulam) at the 2-4 true leaf stage. The quinclorac and florpyrauxifen-benzyl + penoxsulam label rates were 375 and 65 g ai ha−1 respectively (PPPD, 2022). Herbicides that were dissolved in tap water were applied using a motorized backpack sprayer equipped with two flat-fan nozzles mounted on a hand-held boom (Teejet XR11002) and calibrated to deliver 195 L ha−1. Sprayed seedlings were kept in the spraying place for 1 day after treatment. The seedlings were moved to the growth chamber and irrigated using tap water as needed. The seedlings were cut from ground level at 28 DAT and stored in a drying oven at 60 °C for 48 h to determine the above ground dry weight.

Field experiments

Two field experiments were conducted at Bilecik Seyh Edebali University Aşağıköy Agricultural Application and Research Centre (AAARC) in 2021 and 2022 to determine the response of sunflower cultivars to sub-lethal rates of florpyrauxifen-benzyl + penoxsulam and quinclorac under non-irrigated conditions. The soil texture at AARC was silty clay with 2.4% organic matter, pH 8.13. Field studies were arranged within a randomized complete block design with four replications. Conventional sunflower cultivars, Bosfora and Tunca, were sown to 70 cm spacing in plots consisting of four parallel rows on April 11, 2021, and May 02, 2022, respectively. Di-ammonium phosphate was applied at the time of sowing at 80 kg ha−1. In this study, the temperature and rainfall in 2021 and 2022 were close to long-term averages (12.1 and 12.7 °C and 449.2 and 478.9 mm, respectively).

Florochloridone was applied prior to emergence at a 700 g ai ha−1 rate in 2021, while pendimethalin was used two days after sowing at a 1.350 g ai ha−1 rate to control weeds in 2022. Plots consisted of 4 70-cm-spaced rows 10 m long. Two crop rows between the herbicide-applied plots were left as alleys to avoid contamination between the plots.

Quinclorac and florpyrauxifen-benzyl + penoxsulam rates were 11.72, 23.44, and 46.88 g ai ha−1 and 2.03, 4.06, and 8.13 g ai ha−1 respectively, which were equivalent to 3.125, 6.250, and 12.5% of the recommended use rates (PPPD, 2022). The aforementioned rates of herbicides were applied when the sunflower reached 8-10 true leaves. Sunflower injury caused by these herbicide rates was recorded at 7, 14, and 28 DAT based on a scale of 0–100%, where 0% indicated no impact of herbicides, while 100% represented complete plant death. Plants were harvested for yield at the ripening stage. At harvest, five plants from the middle two rows of each plot were randomly selected and harvested by hand on September 05, 2021, and October 04, 2022. The heights of the selected plants, sunflower head diameter (SHD), 1000-seed weight (OTSW), and yield were measured (Serim & Maden, 2014).

Statistical analysis

The data from the dose-response study was evaluated using a nonlinear regression model in R statistical software (RStudio Team, 2023). The DRC package was used to calculate the dose-response curve and parameters with a four-parameter log–logistic model (Equation (1)). (1) Y=c+d−c1+ expblogx−logGR50

where y represents seedling dry matter at herbicide treatment rate x; b, c, d, and GR50 represent slope, lower limit, upper limit, and herbicide rate that reduced seedling dry matter by 50%, respectively.

The data from the field experiments were analyzed with analysis of variance for each herbicide and cultivar, and mean separation was performed with Fisher’s protected least significant difference test at the 5% level of probability. The agricolae package (Mendiburu & Yaseen, 2020) was used in the R statistical software program (RStudio Team, 2023). The Pearson correlation test was used to find a relationship between quinclorac injuries measured at 4 separate times and the yield (and yield components). The test was not performed for FBP because higher rates of FBP killed sunflowers before harvest.

Results and Discussion

Growth chamber experiment

The sunflower cultivar ‘Tunca’ was severely injured when exposed to low rates of florpyrauxifen-benzyl + penoxsulam (Fig. 1A). However, the lowest rate of herbicide unexpectedly increased shoot length, probably due to the hormetic impact of auxinic herbicides, similar to what was previously reported by Mudge et al. (2021). Overall, shoot lengths decreased as the rates increased. The impact of the highest four rates of herbicide was the most destructive, and the growing points of cultivar ‘Bosfora’ were completely killed at these rates at 28 DAT (Fig. 1B). More than half of the leaf area of seedlings exposed to these rates became necrotic. The response of cultivar Bosfora to FBP was similar to that of cultivar Tunca, except at rates higher than 1.02 g ai ha−1 where it was more injured (Figs. 1A and 1B).

Figure 1 Response of Tunca (A and C) and Bosfora (B and D) cultivars to florpyrauxifen-benzyl + penoxsulam (A and B) and quinclorac (C and D) rates in the growth chamber.

Injury from quinclorac in both cultivars increased over time, with a slight and gradual increase in sunflower injury with rate. By 28 DAT, the highest sunflower injury was caused by the highest rate of quinclorac (Figs. 1C and 1D). The response of these sunflower cultivars to lower rates of quinclorac was very similar; however, cultivar Bosfora was slightly more sensitive to quinclorac than cultivar Tunca at higher rates. As opposed to FBP, sunflower plants of both cultivars treated with quinclorac still had live growing points, and necrotic areas on the seedling leaves were relatively limited even at the highest rates. At the lowest rates, FBP resulted in no damage to these cultivars, while at the lowest dose of quinclorac growth reduction was still observed.

The quantitative response of the cultivar Tunca and cultivar Bosfora to FBP and quinclorac was evaluated via a dose-response assay using a log–logistic model. The GR50 values of FBP for the cultivars Tunca and Bosfora were 1.07 and 0.75 g ai ha−1, respectively (Table 1), nearly 2.9% and 2% of the recommended rate of FBP (65 g ai ha−1). The GR50 values of quinclorac for sunflower cultivars were 14.16 and 7.56 g ai ha−1, 3,8% and 2% of the recommended rate of quinclorac (375 g ai ha−1). Similar to the visual herbicidal impact of FBP and quinclorac, the results show that FBP was slightly more injurious than quinclorac to these cultivars (Fig. 2).

Table 1 Nonlinear regression parameters of florpyrauxifen-benzyl + penoxsulam and quincloraca.

Herbicide	bbosfora	btunca	c	d	GR50bosfora	GR50tunca	Comp	Sig.	
FBP	3.22	1.66	0.39	3.31	0.75	1.07	0.70	0.035	
Quinclorac	1.91	4.34	1.57	3.24	7.56	14.16	0.53	0.007	
Notes.

a Abbreviations: FBP, Florpyrauxifen-benzyl + penoxsulam; b, slope of the curve at GR50; d, upper limit; GR50, herbicide rate that reduced seedling dry matter by 50%; Comp, comparison rate (GR50bosfora/GR50tunca); Sig, significance (P < 0.05).

Figure 2 Dose-response curves of florpyrauxifen-benzyl + penoxsulam (left) and quinclorac (right) applied to cultivar Bosfora and cultivar Tuna.

Florpyrauxifen-benzyl + penoxsulam rate for Tunca, y = 0.39 + (2.92 / (1 + (exp(1.66(log (Dose) – log (1.07)))))); florpyrauxifen-benzyl + penoxsulam rate for Bosfora, y = 0.39 + (2.92 / (1 + (exp(3.22(log (Dose) – log (0.75)))))); quinclorac rate for Tunca, y = 1.57 + (1.67 / (1 + (exp(4.34(log (Dose) – log (14.16)))))); quinclorac rate for Bosfora, y = 1.57 + (1.67 / (1 + (exp(1.91(log (Dose) – log (7.56)))))).

Field experiment

Crop injury

Sunflower crops exposed to FBP and quinclorac responded to low-dose treatment, especially the leaves. FBP resulted in typical auxin symptoms, such as parallel veins, cupping, twisting, chlorosis, and distortion (Fig. 3). Quinclorac also caused these symptoms, except it did not demonstrate distortion (Fig. 4). The severity of symptoms was higher in cultivar Bosfora than in cultivar Tunca and increased as the rates increased. Stunting became apparent at 28 DAT for both cultivars. The highest rate of FBP prevented the establishment of sunflower heads in both cultivars (Figs. 5B and 5D).

Figure 3 Impact of florpyrauxifen-benzyl + penoxsulam on sunflower (cultivar Bosfora) at 28 DAT in 2022 (left, lowest drift rate; middle, moderate drift rate; right, highest drift rate).

Figure 4 Impact of quinchlorac on sunflower (cultivar Bosfora) at 28 DAT in 2022 (left, lowest drift rate; middle, moderate drift rate; right, highest drift rate).

Figure 5 Efficacy of drift rates of florpyrauxifen-benzyl + penoxsulam (B and D) and quinclorac (A and C) on heads of Tunca (A and B) and Bosfora (C and D) cultivars (X: recommended rate).

*The recommended rate (X) of florpyrauxifen-benzyl + penoxsulam and quinclorac were 65 and 375 g ai ha−1, respectively.

Quinclorac and FBP injury was low at 7 DAT, and no difference was found between cultivar Tunca and cultivar Bosfora (Figs. 6A and 6B). The response of sunflower cultivar to FBP increased as the rates rose and reached 75–77.5% at 8.13 g ai ha−1 at 7 DAT. Similar to the growth chamber study, quinclorac injury in cultivar Bosfora (13.75–33.75%) and cultivar Tunca (17.5–30%) was lower than those of FBP. At 14 DAT, the response of the cultivars to the herbicides was generally similar to that at 7 DAT. Sunflower injury due to FBP increased with the increase in herbicide rates from 2.03 to 8.13 g ai ha−1 at 14 DAT. Cultivar Bosfora exposure to quinclorac at 11.72, 23.44, and 46.88 g ai ha−1resulted in 15, 22.5, and 38.75% injury, respectively, while cultivar Tunca was less sensitive to these rates at 14 DAT.

Figure 6 Florpyrauxifen-benzyl + penoxsulam (FBP) and quinclorac (Q) injury on the sunflower cultivar Bosfora (A) and cultivar Tunca (B) at 7, 14, 28 DAT, and harvest (%).

1The recommended rate (X) of FBP and quinclorac were 65 and 375 g ai ha−1, respectively.

Sunflower injury at the highest rate of FBP was 100% by 28 DAT while the lowest FBP rate resulted in sunflower injury of 76.25–88.75% (Figs. 6A and 6B). Differences between sunflower cultivars became more apparent at 28 DAT. Although the phytotoxicity of quinclorac increased at 28 DAT, crop injury caused by the herbicide was nearly half that of FBP.

Visible injury rates reached the greatest level at harvest. Sunflower injury at 2.03 g ai ha−1 FBP was higher than >90% for cultivar Bosfora and >85% for cultivar Tunca. The highest FBP rate led to complete death of both sunflower cultivars. The injury increased in severity as the quinclorac rate increased in both cultivars. The injury on cultivar Bosfora and cultivar Tunca ranged from 42.5–57.5% and from 36.25–48.75%, respectively. The severity of injury due to quinclorac was limited to at 7 DAT because injury assessment was only performed on leaves, while this influence was more apparent in evaluations of other crop parameters, including plant height, stem structure, size of flower bud and head, were included in the evaluations. Compared to quinclorac, FBP was more phytotoxic to both cultivars, and this destructive impact was observed even from the first assessments at 7 DAT.

The injury rate of auxin herbicides on sensitive row crops has been reported in some studies. For instance, Schroeder, Cole & Dexter (1979) stated that 2,4-D and dicamba resulted in tremendous injury to sunflowers. In another study, Snipes, Street & Mueller (1992) reported that 17 g ha−1 or higher rates of quinclorac may injure cotton when applied at the cotyledon stage, and cotton injury increased with increasing herbicide rate, especially during late-stage applications, such as at the pin-head square stage. Overall, the cotton injury rate reached 59% at 140 g ha−1 in their study, and sunflower injury caused by quinclorac at 46.88 ai g ha−1 was 52-61.25% in our study. A study conducted by Lovelace et al. (2007) revealed that tomato was also among quinclorac-sensitive crops, and crop injury caused by quinclorac applied at 42 g ai ha−1 was 45% at 49 DAT. In our study, the third evaluation was done at 55 DAT, and crop injuries were 48.75–57.5% when herbicide was applied at 46.88 g ai ha−1. These data are consistent with the work of Lovelace et al. (2007). Comparing these results to those of Snipes, Street & Mueller (1992) and Lovelace et al. (2007), sunflower can be classified as a more sensitive crop than cotton.

Florpyrauxifen-benzyl is a relatively new herbicide on the market; therefore, few drift studies are available in the literature. FBP containing 2.03 g ai ha−1 florpyrauxifen-benzyl + penoxsulam caused 76 and 89% injury at 28 DAT on cultivar Tunca and cultivar Bosfora, respectively. In agreement with the results of our study, Miller & Norsworthy (2018b) stated that 3 g ai ha−1 florpyrauxifen-benzyl applied to sunflowers at the three true-leaf stage resulted in 69% injury, at 28 DAT under greenhouse conditions.

Plant height

The heights of the sunflower cultivars constantly declined as the FBP and quinclorac rates increased. All FBP rates resulted in a significant decrease in the plant height of the sunflower cultivars; The lowest rate of FBP resulted in an 8–9% decrease while the highest resulted in a 32–44% decrease (Fig. 7A). Similarly, the height reduction from quinclorac ranged from 13–14% at the lowest rate to 25–32% at the highest. Similar to the findings of this study, Miller & Norsworthy (2018b) also stated that 3 g ai ha−1 florpyrauxifen-benzyl applied to sunflowers at the three true-leaf stage resulted in a 66% plant height reduction at 28 DAT under greenhouse conditions. The lower plant height observed in our study may be due to the application time of herbicide or penoxsulam in the herbicide mixture.

Figure 7 Yield and yield components of sunflower cultivars in response to FBP and quinclorac (a, Plant height; b, Sunflower head diameter (cm); c, One thousand seed weight (g); Yield (kg ha−1)).

1Means followed by the same letter are not significantly different (P ≤ 0.05). 2FBP, florpyrauxifen-benzyl + penoxsulam. 3The recommended rate (X) of FBP and quinclorac were 65 and 375 g ai ha−1, respectively.

Sunflower head diameter and One thousand seed weight

Increasing FBP and quinclorac rates caused sunflower head diameter (SHD) to decline regardless of the cultivar. The greatest percentage of decline resulting from FBP was recorded at 4.06 g ai ha−1 with 76–77%. At the highest rate of FBP (8.13 g ai ha−1), no flower heads were observed (Figs. 5B and 5D). SHD reduction in cultivar Bosfora and cultivar Tunca from quinclorac ranged from 19–57% and 9–51%, respectively (Fig. 7B). A slight decrease was observed in One thousand seed weight (TSW) compared to plant height and SHD, but this decrease, even at the highest level of herbicide, was no more than 21% (Fig. 7C). Sunflower exposed to higher FBP rates, 4.06 and 8.13 g ai ha−1, could not produce mature seeds. The results showed that TSW was a less susceptible yield component to FBP drift rates than the others.

Yield

Associated with other yield components, significant sunflower yield loss was determined even at lower rates of quinclorac and FBP (Fig. 7D). The highest yield reduction was observed for sunflower cultivars treated with FBP at 4.06 and 8.03 g ai ha−1at 100%. The lowest FBP rate reduced cultivar Bosfora and cultivar Tunca yields by 79.8 and 74.3%, respectively. The yield reduction of cultivar Bosfora from quinclorac rates was 45.1–87.9%, while yield loss of cultivar Tunca caused these rates to range from 16.3–82.3%. Although no study was found in the literature related to the impact of low rates of quinclorac and FBP on sunflower yield and yield components such as SHD and TSW, previous research reported significant reductions in crop growth, yield, and some yield components in other systems. Miller & Norsworthy (2018b) found that soybean yield was reduced 71% when florpyrauxifen-benzyl was applied at 3 g ae ha−1 rate. In another study, Snipes, Street & Mueller (1992) calculated that quinclorac at 50 g ha−1 reduces cotton yield by 25%. On the other side, Lovelace et al. (2007) indicated that quinclorac resulted in yield loss in tomatoes, but the crop may recover itself from the adverse impact of herbicides depending on the herbicide rate and application timing. Our results are consistent with those of the aforementioned studies that lower rates of florpyrauxifen-benzyl were more destructive to crop yields than quinclorac.

Crop cultivars can have various genetic backgrounds depending on the breeding aim; therefore, it is not surprising that they respond differently to abiotic stressors, including drought stress, heat stress, and herbicides. Using herbicide-tolerant crops or less sensitive crop cultivars to herbicide are among the cost-effective and reliable solutions to prevent the injurious impact of herbicide drift on sensitive crops. This practice has been shown in other studies of by France et al. (2022), Zangoueinejad et al. (2021) and Warmund, Ellersieck & Smeda (2022), who showed differences between the tolerance levels of soybean, melon, and tomato cultivars to synthetic auxin herbicides dicamba, 2.4-D, and 2.4-D or dicamba, respectively. The yield data clearly showed that cultivar Bosfora was more sensitive to FBP and quinclorac rates than cultivar Tunca. To reduce the impact of these off-target effects, more cultivars can be screened, and robust cultivars can be selected to reduce the risk of off-target herbicide damage.

Correlation analysis

Remarkably high Pearson correlation coefficients were found between quinclorac injury at 7 or 14 DAT and plant height (Table 2). The negative relations between quinclorac injury at 7 or 14 DAT and SHD, TSW, and/or yield were also high, but their significance was slightly below the relations between quinclorac injury at 7 or 14 DAT and plant height. At 28 DAT, plants treated with quinclorac began to recover from treatment; therefore, quinclorac injury at 28 DAT had a weaker correlation with yield and yield component compared to previous evaluation times. There was a strong positive relationship between plant height and yield, and the importance of the relationship was greater than that of other relationships.

Table 2 Pearson correlation coefficients among the evaluation times after quinclorac application and quinclorac dose, yield, yield components, plant height, sunflower head diameter, and one thousand seed weight.

	Dose	PH	SHD	TSW	Yield	I7DAT	I14DAT	I28DAT	IHarvest	
Dose	1									
PH	−0.88***	1								
SHD	−0.95***	0.87***	1							
TSW	−0.92***	0.75**	0.95***	1						
Yield	−0.90***	0.74**	0.93***	0.95***	1					
I7DAT	0.94***	−0.94***	−0.88***	−0.85***	−0.87***	1				
I14DAT	0.95***	−0.93***	−0.91***	−0.87***	−0.88***	0.99***	1			
I28DAT	0.82**	−0.87***	−0.85***	−0.82**	−0.87***	0.93***	0.93***	1		
IHarvest	0.82**	−0.86***	−0.86***	−0.83**	−0.87***	0.92***	0.93***	1	1	
Notes.

PH plant height

SHD sunflower head diameter

TSW one thousand seed weight

I7DAT injury at 7 DAT

I14DAT injury at 14 DAT

I28DAT injury at 28 DAT

IHarvest injury at harvest

** P < 0.05.

*** P < 0.01.

The correlation analysis performed in this research can be a powerful tool to estimate the injurious impact caused by drift rates of synthetic auxin herbicides on yield and yield components long before harvest. In our study, strong relationships between injury rates and yields (or yield components) were similar to those found in previous studies (Lovelace et al., 2007; Marple, Al-Khatib & Peterson, 2008; Daramola et al., 2023). The ability to model injury rates and yield loss resulting from herbicides provide an opportunity for farmers to decide whether to stop or continue current agricultural practices. If herbicide damage reduces farmer’s income below the total expenses, farmers may wish to change their management in order to remain profitable; therefore, correlation analysis between herbicides and yield can be used as a decision-support tool for farmers.

Conclusions

Each new herbicide introduced into the market has been a new opportunity for rice farmers to control herbicide resistant weeds; however, these new rice synthetic auxin herbicides, FBP and quinclorac, can have destructive impacts on susceptible crops grown nearby such as sunflowers. While both FBP and quinclorac are both synthetic auxins, they work in slightly different ways and therefore have different impacts on crop response. In this study, quinclorac and FBP applied to sunflower cultivars resulted in different injury symptoms and yield losses from different cultivars of sunflowers. Crop injury and yield data clearly showed that cultivar Bosfora was more sensitive to FBP and quinclorac rates than cultivar Tunca, and both cultivars were more sensitive to FBP than quinclorac. The lowest rate of FBP resulted in a 74.3 and 79.8% yield reduction while the higher rates led to a 100% sunflower yield reduction. In our work, we also found that plant height reduction caused by quinclorac at early growth stages may be a valuable indicator to evaluate crop injury and yield loss. Rice growers should be attentive to weather conditions, application parameters, herbicides, herbicide properties, and the safety measurements given by pesticide advisors to prevent drift risk on sunflowers. Moreover, sunflower producers should be careful about the location of sunflower fields prior to sowing and closely communicate with rice producers who use these synthetic auxin herbicides and choose less sensitive sunflower cultivars to synthetic auxin herbicides instead of sensitive ones.

Supplemental Information

Supplemental Information 1 Dataset of Dose-response curves of florpyrauxifen-benzyl + penoxsulam (left) and quinclorac (right) applied to var. Bosfora and var. Tuna

Click here for additional data file.

Supplemental Information 2 Dataset of yield and yield components of sunflower varieties in response to various rates of FBP and quinclorac

Click here for additional data file.

Supplemental Information 3 Injury dataset of Florpyrauxifen-benzyl + penoxsulam and quinclorac on the sunflower var. Bosfora and var. Tunca

Click here for additional data file.

Supplemental Information 4 Quality Parameters Of Irrigation Water

Click here for additional data file.

Additional Information and Declarations

Competing Interests

Author Contributions

Data Availability

Ahmet Tansel Serim is an Academic Editor for PeerJ.

Ahmet Tansel Serim conceived and designed the experiments, performed the experiments, analyzed the data, prepared figures and/or tables, authored or reviewed drafts of the article, and approved the final draft.

Eric L. Patterson analyzed the data, prepared figures and/or tables, authored or reviewed drafts of the article, and approved the final draft.

The following information was supplied regarding data availability:

The raw data is available in the Supplementary Files.

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
