# Peer review of "Response of conventional sunflower cultivars to drift rates of synthetic auxin herbicides"

_PeerJ, doi:10.7717/peerj.16729_

## Round 0.1 · original submission · Major Revisions

Improve title
The abstract is very results-heavy. reduce results from the abstract and make it a bit generalized highlighting the findings.
Improve methodology to make it reproducible.
Foot notes of figures and tables are missing.
Improve discussion write-up.
No need to add citations in the conclusion.

·

Basic reporting

The study is well designed and well explained.

Experimental design

I am satisfied with the experimental design.

Validity of the findings

No comments

Reviewer 2 ·

Basic reporting

no comment

Experimental design

no comment

Validity of the findings

no comment

Additional comments

In my opinion, this article is very well written. It covers the applied side of scientific knowledge. This study concluded with an escape mechanism. I think if a greater number of sunflower genotypes were used then something about differential behavior could be expected. Please share your data about that if something did, even in pilot experiments.

Reviewer 3 ·

Basic reporting

Your manuscript ¨Response of sunflower to drift rates of synthetic auxin herbicides used in rice fields¨ is interesting but needs extensive revisions to be acceptable.
1) Firstly, I would recommend authors to revise the title. Name of different crops in titles is confusing. You can revise it as ¨Response of sunflower to drift rates of synthetic auxin herbicides¨. Abstract
Line 45-46. Focus on sunflower instead the rice.
Introduction
Introduction is well written and organized. However, some latest studies (2022-2023) are suggested to cite.
2ndly, there is need to modify in the light of revised title.
Materials and Methods
1. I would suggest to add a new heading ¨ Herbicide and sunflower varieties¨ and describe the characteristics of herbicides and varieties etc.
2. Why were herbicide rates used in decimals? I would suggest to round off these numeric.
3. Line 171. Mention the solvent of herbicides. How were they dissolved?
4. Line 183. Why was nutrient analysis of soil not added?
5. Line 192. Add, ¨How were plants irrigated in different treatments to avoid cross contamination¨. What was the irrigation interval?
Line 203. Why were herbicides applied at different growth stage as compared to pot experiment?
Line 211-12.Write the complete name of ANNOVA &LSD.¨ It will be more appropriate to add separate heading ¨statistical analysis¨ and describe all the design, analysis in this section.
Results
Add the numeric values of different tested parameters. Avoid just description like increased, decreased and referring the reader to fig. Add important findings quantitatively with their units.
Discussion
Please add the reasoning of each findings in accordance with the similarities and differences of earlier reports.
Conclusion
It needs to be short. Avoid citation of earlier studies. Rather conclude your findings comprehensively.
References
References need to be checked as per journal style. e. g. see line 448-449. Avoid the citation of weblinks if relevant studies are available as journal articles.
Fig 2 & 6: Why is the data not statistical analyzed? SE, comparison among treatments is missing.

Experimental design

Appropriate

Validity of the findings

Good

Additional comments

N/A

·

Basic reporting

1. Most of this paper is well written but there are sections, sentences, or even just words that should be restructured or replaced to better convey the point and be readable. I have given several examples of this in the introduction. This should be done throughout. In particular, the first sentence of the conclusions section should be re-thought. I congratulate the author on conveying the idea of the project well in the introduction.
2. There was sufficient literature review.
3. The figures need better explanation. Provide text below each figure that succinctly describes what the figure is displaying.
4. The discussion and conclusions section may require further thought. The discussion section felt like a continuation of the literature review, at times. I found the conclusion section broad. Think about the purpose of this study and what is most important to convey based on that and the findings.

Experimental design

1. I feel this research is within the scope of the journal and is useful.
2. Based on the paragraph at the end of the introduction, this research appears practically useful for sunflower and rice farmers in, at least, Thrace, and probably other similar areas.
3. Investigation was backed up by literature and thorough.
4. Methods were sufficiently described.

Validity of the findings

1. The specific importance of findings to the purpose of this study could be better expressed in the conclusions. The conclusions are very broad and should be made more specific to the purpose of this study.
Some subjects were mentioned in the conclusion section (UAV's, et al.) that had not been previously mentioned.

Additional comments

The purpose, rationale, and methods of this study are sound. The results section is sound, but could use some re-working on how it is expressed. A better definition of sections would be helpful in the results.
The discussion has a lot of literature review that should be shortened to make the point more quickly.
The conclusions need to be more specific to the study.

I have minor comments on the introduction, materials and methods, and results. The discussion and conclusions would benefit from more thought, revision, and review.

·

Basic reporting

• Condense the introduction section to clearly highlight the problem, research gap, and the study's potential contributions.
• Update the literature review with recent references from 2022-23 and verify the accuracy of the citations.

Experimental design

• Explain the rationale for using a randomized complete block design (RCBD) instead of a completely randomized design in the controlled condition experiment. Provide details on how the blocks were maintained in the climate chamber.
• Clarify how the sprayer was maintained at 175 L ha-1 when spraying only two plants in a pot.
• Include data on the physico-chemical properties of the plant growth medium and the quality parameters of the irrigation water.

Validity of the findings

The results appear to be valid; however, the manner in which they are presented is ambiguous and could benefit from improvement.

Additional comments

Line # 46-47: Consider using the term "agrochemical industry" instead of "agrichemical industry" for better accuracy. Rephrase the sentences for clarity and conciseness.
Line # 51: Revise the phrase "climate room experiment" as it is not clear. Provide a more specific description of the experimental setup.
Line # 61: Instead of writing "11.72, 23.44, 46.88," use "11.72, 23.44, and 46.88" for better readability and consistency.
Line # 69: Specify the crop or vegetable name when mentioning the "prominent growing region."
Line # 69-74: Break down long sentences into shorter ones to improve readability and clarity.
Line # 82: Consider adding some advantages of using herbicides to provide a more comprehensive perspective.
Line # 83-85: Rephrase the sentence for better clarity and coherence.
Line # 85: Specify the locality and timeframe when referring to farmers who were largely unaware of resistance.
Line # 100, 189-193: Make the statement "result in upwards of 93-100% yield reduction" clearer and more understandable for readers.
Line # 188: Define "om" as organic matter in the first mention, followed by the short form "OM" in brackets.
Line # 218: Address the inconsistency between the heading "Results and Discussion" and the subsequent discussion section at line # 317.
Line # 416: Ensure that all references mentioned in the manuscript text are included in the bibliography.

---

## Round 0.2 · Minor Revisions

Please double-check the whole manuscript very carefully, specifically focusing on corrections/suggestions of the Editor, Reviewer-5, correct grammar and formatting mistakes etc.

Write correctly (2 parenthesis together) - of Helianthus annuus (L.) (sunflower) production

Format (space) - withdrift

A lot of such mistakes still exist.

Reviewer 3 ·

Basic reporting

Authors have incorporated revisions and the revised manuscript is acceptable.

Experimental design

Ok

Validity of the findings

Ok

Additional comments

Authors have incorporated revisions and the revised manuscript is acceptable.

·

Basic reporting

Revise with correct grammar

Experimental design

seems okay

Validity of the findings

valid

Additional comments

Regarding the manuscript, it has been reviewed and suggestions have been provided in an annotated PDF file. One of the key recommendations is that the Introduction section is still too lengthy and could benefit from being shortened or restructured to improve its clarity and conciseness.

In addition to the length issue, there are also some grammar corrections that could be made to enhance the overall quality of the manuscript. It is advisable to address these grammatical errors before accepting the manuscript for publication or further evaluation.

By incorporating the suggested changes and addressing the grammar corrections, the manuscript will have a better chance of being well-received by readers and reviewers. This will contribute to its overall effectiveness and impact.

---

## Round 0.3 · Minor Revisions

Make title more clear

Parallel comparison must be made in both experiments like growth chamber and field experiment. In field experiment, efficient of variety was not highlighted, likewise which herbicide was harmless was not highlighted. No indication of varieties efficiency and herbicides effect highlighted in abstract.
Conclusion sentence of abstract is very weak, so improve/add tangible conclusion sentences.

Still in the second revisions you have the mistakes like “loss. respectively.” in the abstract. Replace . with ,
Two styles of writing in same sentence “52-61%” and “85 to 100%”

Please check whole manuscript for such grammar, typo mistake and scientific name italic etc

---

## Round 0.4 · accepted · Accept

I have just one minor edit, which you should address while in production:

Please remove : "(i.e., sublethal rates)" and "(plant height, head diameter, and one-thousand seed weight)" from the abstract.

Otherwise, the article is now Acceptable